# Entanglement asymmetry as a probe of symmetry breaking

Filiberto Ares ®[1] ✉, Sara Murciano[1,2,3] & Pasquale Calabrese[1,4]

Symmetry and symmetry breaking are two pillars of modern quantum physics. Still, quantifying how much a symmetry is broken is an issue that has received little attention. In extended quantum systems, this problem is intrinsically bound to the subsystem of interest. Hence, in this work, we borrow methods from the theory of entanglement in many-body quantum systems to introduce a subsystem measure of symmetry breaking that we dub *entanglement asymmetry*. As a prototypical illustration, we study the entanglement asymmetry in a quantum quench of a spin chain in which an initially broken global $U(1)$ symmetry is restored dynamically. We adapt the quasiparticle picture for entanglement evolution to the analytic determination of the entanglement asymmetry. We find, expectedly, that larger is the subsystem, slower is the restoration, but also the counterintuitive result that more the symmetry is initially broken, faster it is restored, a sort of quantum Mpemba effect, a phenomenon that we show to occur in a large variety of systems.

Symmetries hold a special place in every branch of physics, from relativity to quantum mechanics, passing through gauge/gravity duality and numerical algorithms. It is difficult to identify who was the first in understanding their relevance since the transversal development of the subject is a huge puzzle where different scientists, from Galileo to Noether, gave their own remarkable contributions. Sometimes it happens that, when a parameter reaches a critical value, the lowest energy configuration respecting the symmetry of the theory becomes unstable and new asymmetric lowest energy solutions can be found. This phenomenon does not require an input, whence the name spontaneous symmetry breaking. Other times a symmetry can be explicitly broken, in the sense that the Hamiltonian describing the system contains terms that manifestly break it. The present work fits in this framework: our main goal is to find a tool that measures quantitatively how much a symmetry is broken.

To be more specific, the setup we are interested in is an extended quantum system in a pure state $|\Psi\rangle$, which we divide into two spatial regions $A$ and $B$. The state of $A$ is described by the reduced density matrix $\rho_A = \text{Tr}_B(|\Psi\rangle\langle\Psi|)$. We consider a charge operator $Q$ that generates a global $U(1)$ symmetry group, hence satisfying $Q = Q_A + Q_B$. If $|\Psi\rangle$ is an eigenstate of $Q$, then $[\rho_A, Q_A] = 0$ and $\rho_A$ displays a block-

diagonal structure, with each block corresponding to a charge sector of $Q_A$. Thus the entanglement entropy $S(\rho_A) = -\text{Tr}(\rho_A \log \rho_A)$, which measures how entangled $A$ and $B$ are, can be decomposed into the contributions of each charge sector[1–6] (known as symmetry-resolved entanglement), recently accessed also experimentally[7–10].

Here we consider the opposite situation: a state $|\Psi\rangle$ that breaks the global $U(1)$ symmetry. Therefore, $[\rho_A, Q_A] \neq 0$ and $\rho_A$ is not block-diagonal in the eigenbasis of $Q_A$. The goal of this work is to introduce a quantifier of the symmetry breaking at the level of the subsystem $A$, which is the *entanglement asymmetry* defined as

$$\Delta S_A = S(\rho_{A,Q}) - S(\rho_A). \tag{1}$$

Here $\rho_{A,Q} = \sum_{q \in \mathbb{Z}} \Pi_q \rho_A \Pi_q$, where $\Pi_q$ is the projector onto the eigenspace of $Q_A$ with charge $q \in \mathbb{Z}$. Thus $\rho_{A,Q}$ is block-diagonal in the eigenbasis of $Q_A$. In Fig. 1, we pictorially show how $\rho_{A,Q}$ is obtained from $\rho_A$. A similar quantity, but for the full system, has been recently introduced in ref. 11 to study the inseparability of mixed states with a globally conserved charge.

The entanglement asymmetry (1) satisfies two natural properties to quantify symmetry breaking: (i)$\Delta S_A \geq 0$, because by definition it is equal to the relative entropy between $\rho_A$ and $\rho_{A,Q}$,

[1]SISSA and INFN, via Bonomea 265, 34136 Trieste, Italy. [2]Department of Physics and Institute for Quantum Information and Matter, California Institute of Technology, Pasadena, CA 91125, USA. [3]Walter Burke Institute for Theoretical Physics, California Institute of Technology, Pasadena, CA 91125, USA. [4]The Abdus Salam International Center for Theoretical Physics, Strada Costiera 11, 34151 Trieste, Italy. ✉e-mail: faresase@sissa.it

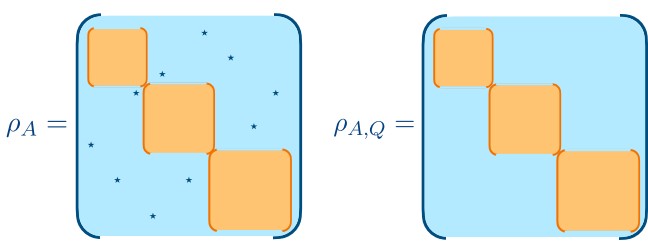

**Fig. 1 | The density matrices $\rho_A$ and $\rho_{A,Q}$ entering in the definitions (1) and (2) of the entanglement asymmetries.** In the eigenbasis of the subsystem charge $Q_A$, $\rho_A$ generically displays off-diagonal elements. Under a projective measurement of $Q_A$, we get $\rho_{A,Q}$, where the off-diagonal blocks are annihilated. The difference $\Delta S_A^{(n)}$ between the entanglement entropies of these matrices is our probe of symmetry breaking.

$\Delta S_A = \mathrm{Tr}[\rho_A(\log\rho_A - \log\rho_{A,Q})]$, which is actually non-negative[12]; (*ii*) $\Delta S_A = 0$ if and only if the state is symmetric since, in this case, $\rho_A$ is block diagonal in the eigenbasis of $Q_A$ and $\rho_{A,Q} = \rho_A$.

## Results

### A replica construction

The entanglement asymmetry can be computed from the moments of the density matrices $\rho_A$ and $\rho_{A,Q}$ by exploiting the replica trick[13,14]. Indeed, simply defining the Rényi entanglement asymmetry as

$$\Delta S_A^{(n)} = \frac{1}{1-n}\left[\log\mathrm{Tr}(\rho_{A,Q}^n) - \log\mathrm{Tr}(\rho_A^n)\right], \quad (2)$$

one has that $\lim_{n\to 1}\Delta S_A^{(n)} = \Delta S_A$. As usual, the advantage of this construction is that, for integer $n$, $\Delta S_A^{(n)}$ can be accessed from (charged) partition functions. Using the Fourier representation of the projector $\Pi_q$, the post-measurement density matrix $\rho_{A,Q}$ can be alternatively written in the form

$$\rho_{A,Q} = \int_{-\pi}^{\pi}\frac{d\alpha}{2\pi}e^{-i\alpha Q_A}\rho_A e^{i\alpha Q_A}, \quad (3)$$

and its moments as

$$\mathrm{Tr}(\rho_{A,Q}^n) = \int_{-\pi}^{\pi}\frac{d\alpha_1\ldots d\alpha_n}{(2\pi)^n}Z_n(\boldsymbol{\alpha}), \quad (4)$$

where $\boldsymbol{\alpha} = \{\alpha_1, \ldots, \alpha_n\}$ and

$$Z_n(\boldsymbol{\alpha}) = \mathrm{Tr}\left[\prod_{j=1}^{n}\rho_A e^{i\alpha_{j,j+1}Q_A}\right], \quad (5)$$

with $\alpha_{ij}\equiv\alpha_i - \alpha_j$ and $\alpha_{n+1} = \alpha_1$. Notice that, if $[\rho_A, Q_A] = 0$, then $Z_n(\boldsymbol{\alpha}) = Z_n(\boldsymbol{O})$, which implies $\mathrm{Tr}(\rho_{A,Q}^n) = \mathrm{Tr}(\rho_A^n)$ and $\Delta S_A^{(n)} = 0$. Furthermore the order of terms in Eq. (5) matters because $[\rho_A, Q_A] \neq 0$. We will refer to $Z_n(\boldsymbol{\alpha})$ as *charged moments* because they are a modification of the similar quantities introduced for the symmetry resolution of entanglement[2].

### Tilted ferromagnet

As warm up, we start with an undergraduate exercise. We consider an infinite spin chain prepared in the tilted ferromagnetic state, i.e. the spins are not aligned with the quantization axis $z$,

$$|\theta; \nearrow\nearrow\cdots\rangle = e^{-i\frac{\theta}{2}\sum_j\sigma_j^y}|\uparrow\uparrow\cdots\rangle. \quad (6)$$

For $\theta \neq \pi m$, $m \in \mathbb{Z}$, this state breaks the $U(1)$ symmetry associated to the conservation of the total transverse magnetization $Q = \frac{1}{2}\sum_j\sigma_j^z$. When $\theta = \pi m$, it corresponds to a fully polarized state in the $z$-direction, for which the transverse magnetization is preserved. The angle $\theta$

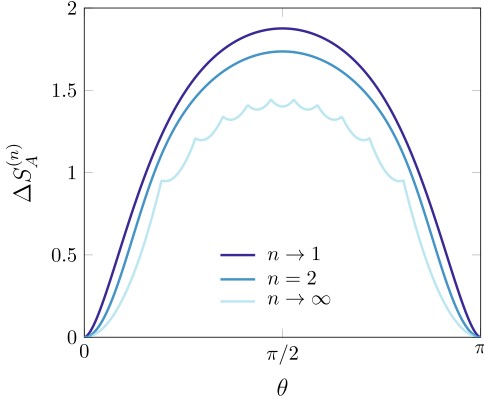

**Fig. 2 | Rényi entanglement asymmetry $\Delta S_A^{(n)}$ for the tilted ferromagnetic state.** We plot the analytic expression of Eq. (8) for this state as a function of the tilting angle $\theta$ for different values of the replica index $n$ and subsystem size $\ell = 10$.

controls how much the state breaks this symmetry and, therefore, the state (6) is an ideal testbed for the entanglement asymmetry, although it is a trivial product state. Let the subsystem $A$ consist of $\ell$ contiguous sites of the chain; then $\Delta S_A = 0$ for $\theta = \pi m$ and $\Delta S_A > 0$ otherwise. Since the state is separable, $\mathrm{Tr}(\rho_A^n) = 1$, and $Z_n(\boldsymbol{\alpha})$ is straightforwardly obtained as

$$Z_n(\boldsymbol{\alpha}) = \prod_{j=1}^{n}\left[i\cos(\theta)\sin\left(\frac{\alpha_{j,j+1}}{2}\right) + \cos\left(\frac{\alpha_{j,j+1}}{2}\right)\right]^{\ell}. \quad (7)$$

Plugging Eq. (7) into the Fourier transform (4), we obtain

$$\Delta S_A^{(n)} = \frac{1}{1-n}\log\left[\cos^{2n\ell}\left(\frac{\theta}{2}\right)\sum_{p=0}^{\ell}\binom{\ell}{p}^n\tan^{2np}\left(\frac{\theta}{2}\right)\right]. \quad (8)$$

In Fig. 2, we plot this entanglement asymmetry as a function of $\theta \in [0, \pi]$. As expected, it vanishes for $\theta = 0, \pi$ while it takes the maximum value at $\theta = \pi/2$, when all the spins point in the $x$ direction and the symmetry is maximally broken. Between these extremal points, $\Delta S_A$ is a monotonic function of $\theta$ (but this is not true for all $n$). For a large interval, it behaves as

$$\Delta S_A^{(n)} = \frac{1}{2}\log\ell + \frac{1}{2}\log\frac{\pi n^{\frac{1}{n-1}}\sin^2\theta}{2} + O(\ell^{-1}). \quad (9)$$

The limit $\theta \to 0$ is not well defined in Eq. (9). Indeed, the limits $\ell \to \infty$ and $\theta \to 0$ do not commute: to recover the symmetry, one should take first $\theta \to 0$ in Eq. (7) and then consider the large interval regime.

### Quench to the XX spin chain

We now analyze the time evolution of the entanglement asymmetry after a quantum quench. We prepare the infinite spin chain in the state

$$|\Psi(0)\rangle = \frac{|\theta; \nearrow\nearrow\cdots\rangle - |-\theta; \nearrow\nearrow\cdots\rangle}{\sqrt{2}}, \quad (10)$$

which is the *cat* version of the symmetry-breaking state in Eq. (6). We then let it evolve

$$|\Psi(t)\rangle = e^{-itH}|\Psi(0)\rangle, \quad (11)$$

with the symmetric XX Hamiltonian ($[H, Q] = 0$)

$$H = -\frac{1}{4}\sum_{j=-\infty}^{\infty}\left[\sigma_j^x\sigma_{j+1}^x + \sigma_j^y\sigma_{j+1}^y\right]. \quad (12)$$

This Hamiltonian is diagonalized via the Jordan-Wigner transformation to fermionic operators followed by a Fourier transform to momentum space[15]. The one-particle dispersion relation is $\epsilon(k) = -\cos(k)$.

## The entanglement asymmetry after the quench

At time $t = 0$, the entanglement asymmetry behaves asymptotically as Eq. (9); for $t > 0$, $\Delta S_A^{(n)}(t)$ is analytically derived in Methods by adapting the *quasiparticle picture* of entanglement dynamics[16–18] to the charged moments (5) and then taking the Fourier transform (4). The resulting curves are plotted in Fig. 3 as a function of $\zeta = t/\ell$ for several values of $\theta$, finding a remarkable agreement with the exact numerical values (symbols). We can also write a very effective closed-form approximation of $\Delta S_A^{(n)}(t)$,

$$\Delta S_A^{(n)}(t) \simeq \frac{\pi^2 b(\zeta)\ell}{24},$$
$$b(\zeta) = \frac{\sin^2\theta}{2} - \int_0^{2\pi}\frac{dk}{2\pi}\min(2\zeta|\epsilon'(k)|,1)\sin^2\Delta_k, \tag{13}$$

which is independent of the replica index $n$ (see Methods for the definition of $\Delta_k$). This approximation becomes *exact* in the limit of large $\zeta$ and its effectiveness, also for not too large $\zeta$, is proven by the inset of Fig. 3.

We now discuss some relevant features of the entanglement asymmetry and show that it encodes a lot of new physics. First, as expected[19,20], $\Delta S_A^{(n)}(t)$ tends to zero for large $\zeta$ (i.e. large $t$) and the $U(1)$ symmetry, broken by the initial state, is restored. This is analytically shown by Eq. (13) that indeed at leading order in large $\zeta$ is

$$\Delta S_A^{(n)}(t) \simeq \frac{\pi}{1152}\left(1 + 8\frac{\cos^2\theta}{\sin^4\theta}\right)\frac{\ell}{\zeta^3}, \tag{14}$$

i.e. it vanishes for large times as $t^{-3}$ for any value of $\theta$. This decay is determined by the quasiparticles with the slowest velocity $|\epsilon'(k)|$, which in this case are those with momentum around $k = 0$ and $\pi$. Another characteristic, following from having a space-time scaling, is that larger subsystems require more time to recover the symmetry, as it is clear from Fig. 3 and Eq. (13): this justifies the significance of the definition of $\Delta S_A^{(n)}$ in terms of $\rho_A$ rather than the full state $|\Psi\rangle$. Finally, a very odd and intriguing feature is that the more the symmetry is initially broken, i.e. the larger $\theta$, the smaller the time to restore it. This is a quantum Mpemba effect[21]: more the system is out of equilibrium, the faster it relaxes. At a qualitative level this is a consequence of the fact that for larger symmetry breaking there is a sharper drop of the (entanglement) asymmetry at short time, see Fig. 3, before the truly asymptotic behavior takes place. Furthermore, we can quantitatively understand the quantum Mpemba effect: from Eq. (14) the prefactor of the $t^{-3}$ decay is a monotonously decreasing function of $\theta$ in $[0, \pi/2]$. Thus the quantum Mpemba effect is not as controversial as its classical version[22]. To the best of our knowledge this awkward effect was not known in the literature, showing the power of the entanglement asymmetry to identify new physics.

## Quantum Mpemba effect

The quantum Mpemba effect is not a prerogative of integrable free systems, such as the XX spin chain, but it turns out to be much more general and robust. To show this, we analyze now a global quantum quench having as initial state the tilted ferromagnetic configuration of Eq. (6) and evolving with the interacting Hamiltonian

$$H = -\frac{1}{4}\sum_{j=1}^{N}\left[\sigma_j^x\sigma_{j+1}^x + \sigma_j^y\sigma_{j+1}^y + \Delta\sigma_j^z\sigma_{j+1}^z\right]$$
$$-\frac{J_2}{4}\sum_{j=1}^{N}\left[\sigma_j^x\sigma_{j+2}^x + \sigma_j^y\sigma_{j+2}^y + \Delta_2\sigma_j^z\sigma_{j+2}^z\right], \tag{15}$$

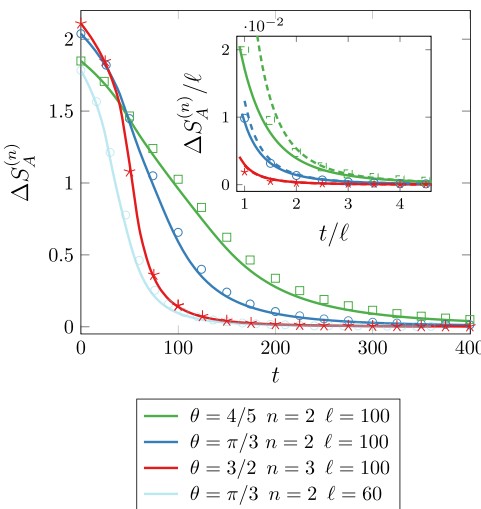

**Fig. 3 | Time evolution of the Rényi entanglement asymmetry $\Delta S_A^{(n)}(t)$ after the quench (11).** The symbols are the exact numerical results for various values of the subsystem length $\ell$, the replica index $n$, and the initial tilting angle $\theta$ (see Methods). The continuous lines are our prediction obtained by plugging the charged moments reported in Methods into (4) and (2). In the inset, we check the asymptotic behavior (13) (full lines) and (14) (dashed) of $\Delta S_A^{(n)}(t)$ for large $t/\ell$. Source data are provided as a Source Data file.

where $N$ is the total number of spins. This Hamiltonian commutes with the transverse magnetization $Q = \frac{1}{2}\sum_j\sigma_j^z$. For $J_2 = 0$, it corresponds to the Heisenberg XXZ spin chain with anisotropy parameter $\Delta$, which is the prototype of all interacting integrable models. For $\Delta = J_2 = 0$, we recover the XX spin chain of the previous paragraphs. For $J_2 \neq 0$, the next nearest neighbor couplings break integrability[23].

The $U(1)$ symmetry is expected to be restored after a generic quench to the Hamiltonian (15)[19]. In fact, at late times, the local stationary behavior is described by a statistical ensemble, corresponding to thermal or generalized Gibbs for chaotic or integrable systems respectively[24–28]. In one dimensional quantum systems, the Mermin-Wagner theorem forbids the spontaneous breaking of a continuous symmetry at finite temperature. In the quench, the finite energy density of the initial state plays the role of an effective temperature, causing in general symmetry restoration (with the exception of very few pathological cases).

In Fig. 4, we plot the time evolution of $\Delta S_A$ after a quench using the Hamiltonian (15) with $N = 10$ spins for different values of the couplings and initial tilting angle $\theta$. In all the cases, the curves have been obtained by applying exact diagonalization. In panels **a** and **b** of Fig. 4, we perform a quench to a periodic XXZ chain ($J_2 = 0$) with interaction $\Delta = 0.4$ (panel **a**) and $\Delta = 3.75$ (panel **b**). In panels **c** and **d** of Fig. 4, the post-quench Hamiltonian contains next nearest neighbor terms ($J_2 = 1$) and, therefore, is non-integrable. Panel **c** corresponds to periodic boundary conditions (PBC) while in panel **d** we consider open boundary conditions (OBC) with the subsystem located at the middle of the chain. In all the plots, the quantum Mpemba effect is clearly visible: the more the symmetry is initially broken, the faster $\Delta S_A(t)$ decays to zero after the quench; this is true, although the finite size of the system causes revivals that prevent us from observing the restoration in a neat way as happens in the thermodynamic limit in Fig. 3.

In conclusion, Fig. 4 shows that quantum Mpemba effect occurs under very general conditions (both for integrable and non-integrable interactions with different boundary conditions), even for (sub)systems of few sites, which makes possible to observe it experimentally in, e.g., ion trap setups.

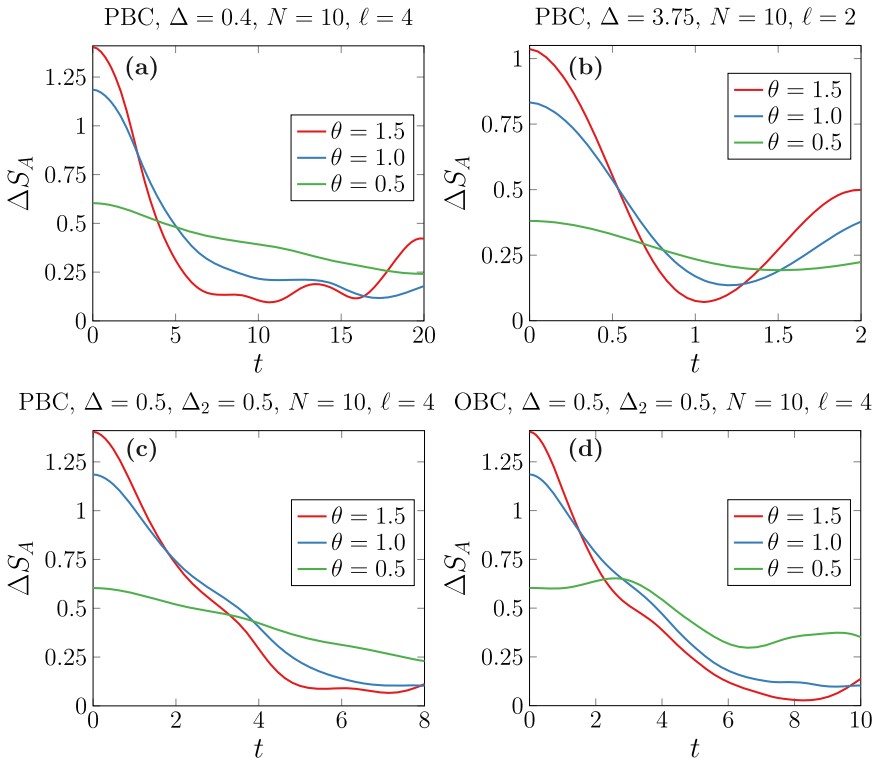

**Fig. 4 | Quantum Mpemba effect in interacting integrable and non-integrable spin chains.** We plot the time evolution of the entanglement asymmetry $\Delta S_A(t)$ after preparing the spin chain at $t = 0$ in the tilted ferromagnetic state of Eq. (6) and performing a sudden quench to the Hamiltonian $H$ given in Eq. (15) with total length $N = 10$. The continuous lines have been obtained via exact diagonalization for different choices of the subsystem length $\ell$, initial tilting angle $\theta$, and of the couplings and boundary conditions in the evolution Hamiltonian $H$. In panels **a** and **b**, $J_2 = 0$, and $H$ corresponds to the XXZ spin chain with anisotropy parameter $\Delta$; in both cases, we take PBC for the chain. In panels **c** and **d**, $J_2 = 1$, and the chain is non-integrable; in panel **c**, we consider PBC while in panel **d** we choose OBC with the subsystem $A$ placed in the middle of the chain. Source data are provided as a Source Data file.

## Discussion

In this work, we introduced the entanglement asymmetry, a probe to study how much a symmetry is broken at the level of subsystems of many-body systems. As an application to show its potential, we have studied its dynamics after a quench from an initial state breaking a $U(1)$ symmetry and evolving with a Hamiltonian preserving it. We showed that the entanglement asymmetry detects neatly all the physical relevant features of the dynamics and in particular the restoration of the symmetry at late times. It also identifies the appearance of an unexpected Mpemba effect, a phenomenon that, as we have seen, happens in many settings that can be studied through the entanglement asymmetry. It is then very important to study other quench protocols (e.g. different initial state, interacting Hamiltonians, etc.) and understand how to modify the quasiparticle description, following e.g. ref. 29, to describe these more general situations.

We can easily imagine many other applications of the entanglement asymmetry. The first one is in equilibrium situations that have been left out here. In this respect, it would be useful to recast the charged moments (5) in terms of twist fields[14,30] within the path-integral approach: this would allow us to explore more complicated situations, e.g. the symmetry breaking from $SU(2)$ to $U(1)$, which are also relevant in high-energy physics[31]. Similarly, our setup can be extended to non-Abelian symmetries[32] to explore, e.g., how the asymptotic behavior of $\Delta S_A$ with the subsystem size of Eq. (9) depends on the symmetry group. Finally, $\Delta S_A^{(n)}(t)$, with $n$ integer $n \geq 2$, can be experimentally accessible by developing a protocol based on the random measurement toolbox[33–35]. This would require the post-selection of data from an experiment like the one in[36], but with an initial state breaking the $U(1)$ symmetry.

## Methods

We provide here the details about the derivation of the numerical and analytical results reported in the Results section.

### Numerical techniques

We choose as initial state the linear combination of Eq. (10), instead of Eq. (6), because, after a Jordan-Wigner transformation, the corresponding reduced density matrix is Gaussian in terms of the fermionic operators $c_j = (c_j^\dagger, c_j)$. We can then use Wick theorem to express $\rho_A(t)$ in terms of the two-point correlation matrix

$$\Gamma_{jj'}(t) = 2 \operatorname{Tr}\left[\rho_A(t) c_j^\dagger c_{j'}\right] - \delta_{jj'}, \tag{16}$$

with $j, j' \in A$[37]. If $A$ is a subsystem of length $\ell$, then $\Gamma(t)$ has dimension $2\ell \times 2\ell$ and entries[38]

$$\Gamma_{jj'}(t) = \int_0^{2\pi} \frac{dk}{2\pi} \mathcal{G}(k,t) e^{-ik(j-j')}, \tag{17}$$

with

$$
\mathcal{G}(k,t) = \begin{pmatrix} \cos \Delta_k & -ie^{-2it\epsilon(k)} \sin \Delta_k \\ ie^{2it\epsilon(k)} \sin \Delta_k & -\cos \Delta_k \end{pmatrix},
$$
$$
\cos \Delta_k = \frac{2\cos(\theta) - (1+\cos^2\theta)\cos(k)}{1+\cos^2\theta - 2\cos(\theta)\cos(k)}. \tag{18}
$$

Under the Jordan-Wigner transformation, the transverse magnetization is mapped to the fermion number operator and $e^{i\alpha Q_A}$ turns out to be Gaussian, too. Therefore, $Z_n(\boldsymbol{\alpha})$ in Eq. (5) is the

trace of the product of Gaussian fermionic operators, $\rho_A$ and $e^{i\alpha_{j,j+1}Q_A}$. Employing their composition properties[39,40], we express $Z_n(\boldsymbol{\alpha})$ as a determinant involving the corresponding correlation matrices, finding

$$Z_n(\boldsymbol{\alpha},t) = \sqrt{\det\left[\left(\frac{I-\Gamma(t)}{2}\right)^n\left(I+\prod_{j=1}^n W_j(t)\right)\right]}, \quad (19)$$

with $W_j(t) = (I+\Gamma(t))(I-\Gamma(t))^{-1}e^{i\alpha_{j,j+1}n_A}$ and $n_A$ is a diagonal matrix with $(n_A)_{2j,2j} = 1$, $(n_A)_{2j-1,2j-1} = -1$, $j = 1,\cdots,\ell$. We use Eq. (19) to numerically compute the time evolution of the Rényi entanglement asymmetry $\Delta S_A^{(n)}(t)$ in Fig. 3 and test the analytical predictions presented in this work.

## Analytic computation

After the quench, the natural ballistic regime is the scaling limit $t,\ell \to \infty$ with $\zeta = t/\ell$ fixed[38,41], in which we find

$$Z_n(\boldsymbol{\alpha},t) = Z_n(\mathbf{0},t)e^{\ell(A(\boldsymbol{\alpha})+B(\boldsymbol{\alpha},\zeta))}, \quad (20)$$

where the functions $A(\boldsymbol{\alpha})$ and $B(\boldsymbol{\alpha},\zeta)$ read, respectively,

$$A(\boldsymbol{\alpha}) = \int_0^{2\pi}\frac{dk}{2\pi}\log\prod_{j=1}^n f(e^{i\Delta_k},\alpha_{j,j+1}),$$

$$B(\boldsymbol{\alpha},\zeta) = -\int_0^{2\pi}\frac{dk}{2\pi}\min(2\zeta|\epsilon'(k)|,1)\log\prod_{j=1}^n f(e^{i\Delta_k},\alpha_{j,j+1}), \quad (21)$$

and $f(\lambda,\alpha)$ is defined as

$$f(\lambda,\alpha) = i\lambda\sin\left(\frac{\alpha}{2}\right)+\cos\left(\frac{\alpha}{2}\right). \quad (22)$$

Notice that in Eq. (21) there is a factorization in the replica space indexed by $j$. This cumbersome expression does not come out of a magician hat, but from the quasiparticle picture[16–18]: the time evolution of the entanglement is given by the pairs of entangled excitations shared by $A$ and $B$ that are created after the quench and propagate ballistically with momentum $\pm k$. Let us explain how to apply this idea to deduce Eq. (21). According to refs. 19,20, in the quench protocol analyzed here, the $U(1)$ symmetry is restored in the large time limit, i.e. $\Delta S_A^{(n)}(t) \to 0$. Therefore, $Z_n(\boldsymbol{\alpha},t)$ has to tend to $Z_n(\mathbf{0},t)$, which implies $B(\boldsymbol{\alpha},\zeta) \to -A(\boldsymbol{\alpha})$ as $\zeta \to \infty$. At time $t = 0$, plugging the initial state of Eq. (10) in the definition of the charged moments (5), we obtain that, for large $\ell$, $Z_n(\boldsymbol{\alpha},0) \sim e^{A(\boldsymbol{\alpha})\ell}/2^{n-1}$ with

$$A(\boldsymbol{\alpha}) = \log\prod_{j=1}^n e^{i\sigma_j/2}f(\cos(\theta),\alpha_{j,j+1}-\sigma_j), \quad (23)$$

where $\sigma_j = 0$ if $|\alpha_{j,j+1}| \le \pi/2$ and $\sigma_j = \pi$ otherwise. Considering Eq. (23), we notice that $Z_n(\boldsymbol{\alpha},0)$ factorizes into

$$Z_n(\boldsymbol{\alpha},0) \sim 2\prod_{j=1}^n \frac{e^{i\sigma_j/2}}{2}\text{Tr}(\rho_A(0)e^{i(\alpha_{j,j+1}-\sigma_j)Q_A}). \quad (24)$$

The expectation value $\text{Tr}(\rho_A(0)e^{i\alpha Q_A})$ is the full counting statistics (FCS) of the transverse magnetization in the subsystem $A$. We can now take advantage of the fact that $|\Psi(0)\rangle$ is also the ground state of a XY spin chain to exploit the knowledge of the FCS in that system[42–46] (the corresponding parameters $h$, $\gamma$ of the XY chain are given by $\gamma^2+h^2 = 1$ and $\cos^2\theta = (1-\gamma)/(1+\gamma)$). In particular, employing the results of ref. 46, we can rewrite $A(\boldsymbol{\alpha})$ in Eq. (23) as an integral in momentum

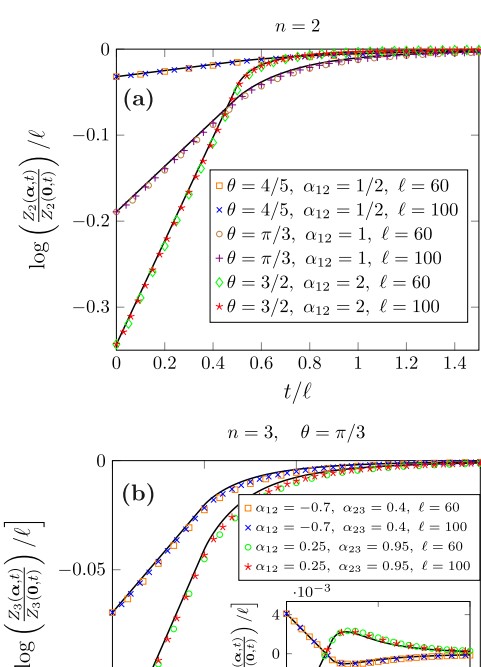

**Fig. 5 | Time evolution of the charged moments $Z_n(\alpha,t)$ after the quench (11).** We plot them as a function of $t/\ell$ for the replica indices $n = 2$ (panel **a**) and $n = 3$ (panel **b**) and several values of the initial tilting angle $\theta$, the subsystem size $\ell$, and the phases $\alpha_{j,j+1}$. The symbols were obtained numerically using Eq. (19) and the continuous lines correspond to the analytic prediction (20). Source data are provided as a Source Data file.

space

$$A(\boldsymbol{\alpha}) = -B(\boldsymbol{\alpha},\zeta \to \infty) = \int_0^{2\pi}\frac{dk}{2\pi}\log\prod_{j=1}^n f(e^{i\Delta_k},\alpha_{j,j+1}). \quad (25)$$

Now, using the quasiparticle picture, the integrand in Eq. (25) can be interpreted as the contribution to $B(\boldsymbol{\alpha},\zeta)$ from each entangled excitation of momentum $k$ created after the quench. Since they propagate with velocity $|\epsilon'(k)|$, the number of these pairs shared between $A$ and its complement at time $t$ is determined by $\min(2t|\epsilon'(k)|,\ell)$. Combining these two ingredients, we get Eq. (20). This approach makes also clear the crucial role that entanglement plays in the restoration of the symmetry. Likely this expression can be rigorously derived by properly adapting the calculations for the symmetry-resolved entanglement[47,48], but this is far beyond the scope of this work. In Fig. 5, we check Eq. (20) against exact numerical computations performed using Eq. (19) for different values of $n$, $\theta$, and $\boldsymbol{\alpha}$, finding a remarkable agreement: note that Eq. (20) is exact for $\ell \to \infty$ and the points are closer to the curves for larger $\ell$. Finally, when in Eq. (20) $A(\boldsymbol{\alpha})+B(\boldsymbol{\alpha},\zeta)$ is close to zero, the Fourier transform (4) can be done analytically and we obtain the approximation for the entanglement asymmetry in Eq. (13).

## Data availability

The data that support the plots within this paper are provided in the Source Data file. Source data are provided with this paper.

## Code availability

The computer codes used to generate the results that are reported in this paper are available from the authors upon reasonable request.

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

## Acknowledgements

The authors thank Jerome Dubail, Viktor Eisler, Maurizio Fagotti, Israel Klich, Lorenzo Piroli, Eric Vernier, and Lenart Zadnik for useful discussions. All the authors acknowledge support from ERC under Consolidator

grant number 771536 (NEMO). SM thanks support from Caltech Institute for Quantum Information and Matter and the Walter Burke Institute for Theoretical Physics at Caltech.

## Author contributions

F.A., S.M., and P.C. contributed to the numerical and analytic computations, the interpretation of the results, developing of the theory and the writing of the manuscript.

## Competing interests

The authors declare no competing interests.
