## [Peer Review File · Nature Communications]

REVIEWERS' COMMENTS

Reviewer #2 (Remarks to the Author):

The major revision in this submission is the extension to the next-nearest-neighbor XXZ model in Eq. (15). The additional numerical evidences in Fig. 4 show that Mpemba effects exist in non-integrable models, not an edge effect (different boundary conditions) and not related to the groundstate phases (varying Δ). The authors also cite Mermin-Wagner theorem to motivate its generality. By and large, the revision have cleared my concerns. I recommend acceptance.

I have one more technical question that hopefully can improve understanding of ΔS_A in free model.

In Methods, it looks like the entanglement asymmetry calculation of the XY model can be factorized into individual modes, where each mode has a function similar to Eq. (7) in the warm-up problem. Is the ΔS_A additive for each mode, so that we can integrate the quasi-particle contribution just like the entanglement itself? I'm trying to see the $1/t^3$ decay scaling in a simple way.

Reviewer #3 (Remarks to the Author):

I would like to thank the authors for the reply and improvement. However, I still believe the significance of the results does not satisfy the criteria for publication in this journal.

Response to Reviewers

Reviewer 2 (Remarks to the Author): *The major revision in this submission is the extension to the next-nearest-neighbor XXZ model in Eq. (15). The additional numerical evidences in Fig. 4 show that Mpemba effects exist in non-integrable models, not an edge effect (different boundary conditions) and not related to the groundstate phases (varying Delta). The authors also cite Mermin-Wagner theorem to motivate its generality. By and large, the revision have cleared my concerns. I recommend acceptance.*

I have one more technical question that hopefully can improve understanding of ΔS_A in free model. In Methods, it looks like the entanglement asymmetry calculation of the XY model can be factorized into individual modes, where each mode has a function (22) similar to Eq. (7) in the warm-up problem. Is the ΔS_A additive for each mode, so that we can integrate the quasi-particle contribution just like the entanglement itself? I'm trying to see the $1/t^3$ decay scaling in a simple way.

We thank the referee for the positive comments about the revised manuscript and the recommendation for acceptance. With respect to their question, the factorization of the charged moments $Z_n(\boldsymbol{\alpha})$ in Eqs. (7) and (22) refers to the number n of copies of the projected reduced density matrix $\rho_{A,Q}$ that we take in the Rényi entanglement asymmetry $\Delta S_A^{(n)}$. Therefore, the factorization is not related to the modes k of the quasi-particles. To obtain $\Delta S_A^{(n)}$, one has to perform the multi-dimensional Fourier transform (4) of the charged moments. Thanks to their factorization in the n copies of $\rho_{A,Q}$, the Fourier transform can be done analytically at large times and the entanglement asymmetry is given by Eq. (13); observe that, in this regime, $\Delta S_A^{(n)}$ is actually additive in the quasi-particle modes k and we can integrate their contribution just like we do in the entanglement entropy. Analogously to the entanglement entropy, the leading order term for large t , in our case $1/t^3$, is determined by the quasi-particles with the slowest velocity $|\epsilon'(k)|$, for free fermions they are those with momentum around $k = 0$ and π since $|\epsilon'(k)| = |\sin(k)|$. We are very grateful to the referee for raising this valuable question, and we have added in the final revised version a comment about this after Eq. (14).